# Exclusive Breastfeeding at Discharge in Regional New South Wales, Australia: The Role of Antenatal Care (2011–2020)

**DOI:** 10.3390/ijerph20126135

**Published:** 2023-06-15

**Authors:** Emma Woolley, Gretchen Buck, Jackie Jackson, Rebekah Bowman, Louise Fox, Shirlena Gallagher, Malindey Sorrell, Pramesh Raj Ghimire

**Affiliations:** 1Integrated Care and Allied Health, Southern New South Wales Local Health District, Queanbeyan, NSW 2620, Australiapramesh.ghimire@health.nsw.gov.au (P.R.G.); 2Nursing and Midwifery, Southern New South Wales Local Health District, Queanbeyan, NSW 2620, Australia; 3People and Culture, Southern New South Wales Local Health District, Queanbeyan, NSW 2620, Australia; 4The Family Place, Moruya, NSW 2537, Australia

**Keywords:** antenatal care, perinatal care, breastfeeding, exclusive breastfeeding, maternal health, New South Wales, Australia

## Abstract

Increasing the number of infants exclusively breastfeeding on discharge from the hospital after birth is a key goal of breastfeeding policy in New South Wales (NSW), Australia. Despite consistent efforts, exclusive breastfeeding on discharge rates have declined over the past decade. Using pooled data from the New South Wales Perinatal Data Collection from 2011 to 2020, we examined the association between antenatal care (ANC) and exclusive breastfeeding at discharge from birth admission outcomes for mother–baby dyads in Southern New South Wales Local Health District (SNSWLHD). Our study confirmed that exclusive breastfeeding rates in SNSWLHD have declined over the past decade, providing local evidence to support action. Late entry to ANC and a failure to attend the recommended number of ANC visits were important predictors of a lower rate of exclusive breastfeeding on discharge. Improving accessibility to ANC visits for rural and regional mothers has potential to positively impact breastfeeding rates in SNSWLHD. We suggest that wider implementation of caseload midwifery models may have a positive impact on breastfeeding outcomes in the region for all mother–baby dyads, but particularly for Aboriginal mothers and infants, younger mothers and mothers experiencing disadvantage.

## 1. Introduction

Breastfeeding is one of the most important components of early childhood nutrition, providing essential nutrients and immunological protection to infants [1]. Despite these benefits, many Australian women struggle to establish and maintain a successful breastfeeding relationship with their baby [2].

New South Wales (NSW) is one of eight states and territories in Australia. In NSW, 15 Local Health Districts manage public hospitals and health facilities, and provide a range of public healthcare services to defined geographical areas across the state. The Southern NSW Local Health District (SNSWLHD), the subject of the present study, covers a large area of the southeastern corner of the state, a total of 44,537 square kilometres, and services a largely rural population of 219,100 people [3]. The Southern NSW Local Health District services just 2.68% of the population of NSW, despite making up 5.56% of the land area [3].

Globally, Indigenous women experience poorer maternal health and maternity mortality than their non-Indigenous counterparts [4]. Commensurate with the experiences of Indigenous women in lower- and middle-income countries, there is an alarmingly wide gap in the prevalence of some key maternal health indicators for Aboriginal Australian mothers and babies compared to non-Aboriginal mothers and babies, including higher rates of low birthweight, preterm birth and smoking in pregnancy [5]. Note that in NSW Health, the term ‘Aboriginal’ is used to describe the original people of Australia and their descendants, in preference to ‘Aboriginal and Torres Strait Islander’, in recognition that Aboriginal people are the original inhabitants of our state.

To promote perinatal health outcomes, the Australian Pregnancy Care Guidelines outline the recommended number and timing of antenatal care (ANC) visits for expectant mothers [6]. These guidelines are intended to ensure that mothers receive comprehensive and quality care during pregnancy, with the goal of promoting optimal health outcomes for both mother and baby [6]. The current guidelines suggest a minimum of ten ANC visits for first-time mothers with an uncomplicated pregnancy (seven visits for subsequent uncomplicated pregnancies), with the first visit taking place in the first trimester (in the first ten weeks of pregnancy) and the last visit occurring close to the expected date of birth [6].

In Australia, antenatal visits are structured around specific content, based on women’s needs [6]. Assessments and tests are scheduled within these visits to ensure antenatal care is delivered systematically, and minimising inconvenience to the woman. In the Southern NSW Local Health District, ANC is provided by public midwives, by the woman’s general practitioner or by an obstetrician, with shared care arrangements being common. The visits occur at the hospital, in a community health centre or, for some women, at home. Following the first ANC visit, each subsequent visit discusses infant feeding.

Antenatal care aims to provide an opportunity for healthcare providers to educate and support expectant mothers in preparing for successful breastfeeding. The Pregnancy Care Guidelines outline when and how breastfeeding information and support should be provided in the context of antenatal visits; however, in practice, the content of the visits is at the clinician’s discretion and depends on the woman’s individual needs at the time of the visit, and other clinical issues may take priority.

Despite well-established pregnancy care guidelines, and a policy directive [7] to promote, protect and support exclusive breastfeeding on discharge, New South Wales health data suggested that exclusive breastfeeding on discharge has decreased from 82.0% in 2011 to 69.1% in 2020 [5].

During 2020, significant healthcare service disruption occurred due to the COVID-19 pandemic to reduce the risk of infection for pregnant women and staff. Shutdowns and service disruptions affected the ability of women to attend face-to-face antenatal care visits, including in the Southern NSW Local Health District. The impact of COVID-19 itself was limited in the Southern NSW Local Health District during 2020, with just 67 confirmed cases, including 54 cases acquired overseas. However, international studies have raised concerns that associated service disruptions could contribute to changes in pregnancy outcomes, and may be further explored once subsequent data are available through the National Perinatal Data Collection [8,9].

A Cochrane review found that mothers who attend more ANC visits are more likely to initiate breastfeeding and continue breastfeeding for a longer duration [10]. This is likely due to several factors, including increased access to information and resources, better identification of potential breastfeeding problems and increased support from healthcare providers [11,12]. Additionally, mothers who attend more antenatal care visits are more likely to receive prenatal education on the benefits of breastfeeding and how to successfully breastfeed, which can increase their confidence and ability to breastfeed [13,14]. The existing literature on exclusive breastfeeding on discharge in Australia is primarily focused on various sociodemographic factors [15,16,17], and research identifying the role of antenatal care in exclusive breastfeeding is limited, particularly in regional settings.

This study aimed to examine trends and investigate the role of antenatal care in exclusive breastfeeding at discharge in the Southern NSW Local Health District by utilising pooled perinatal data for the period 2011–2020. The findings help us to understand the disparity of exclusive breastfeeding on discharge in the Southern NSW Local Health District and provide insight into the cultural and societal factors that should be considered in the design and implementation of tailored perinatal care strategies to promote, protect and support exclusive breastfeeding on discharge. This research provides valuable insights into the effectiveness of current practice in promoting exclusive breastfeeding in Southern NSW and suggests potential improvements to policy and practice to better support mothers in achieving their breastfeeding goals.

## 2. Materials and Methods

### 2.1. Data Source and Sample Composition

The data source for this study was the NSW Perinatal Data Collection (PDC), a statewide population-based surveillance system that collects information around demographic and health indicators, such as timing and number of antenatal care visits, exclusive breastfeeding at discharge, smoking during pregnancy and birthweight, to inform evidence-based health policy and programs aimed to improve maternal and newborns health outcomes in NSW. Each state and territory in Australia has its own system for collecting perinatal data.

This study utilised a cohort of mothers (N = 13,168) who gave birth between 2011 and 2020 in five maternity services of the Southern NSW Local Health District. The details of the sample selected for this study are described in Figure 1; the details of the study settings can be found elsewhere [18].

### 2.2. Outcome Variable

Following birth, information on types of feeding at discharge (breastmilk, expressed breastmilk or infant formula) is routinely collected in an electronic medical record system (eMaternity). The outcome variable for this study was exclusive breastfeeding at discharge. The outcome variable was considered to be dichotomous (1 = Yes, if the baby was feeding by breast or on expressed milk, and 0 = No, if the baby was feeding on infant formula or infant formula and breastmilk/expressed breastmilk). Because of the lack of data, this study was not able to include a complete time frame for exclusive breastfeeding (0–6 months) as an outcome measure.

### 2.3. Exposure Variables

The exposure variables of this study were the number and the timing of first antenatal care (ANC). The Australian Pregnancy Guideline recommends a schedule of ten ANC visits for first pregnancy without complications, and seven visits for subsequent uncomplicated pregnancy. Therefore, the first exposure variable of this study was the recommended number of ANC visits, which was categorised as ‘1’ if women received a recommended number of ANC visits or ‘2’ if women did not receive a recommended number of ANC visits. The timing of first ANC was categorised into three groups (1 = first trimester, 2 = second trimester and 3 = third trimester).

### 2.4. Covariates

The use of covariates in this study was based on the existing literature on the initiation of exclusive breastfeeding [15,16,17] and information available in the data file utilised for this study. We adopted Andersen’s expanded behavioural model of health service use; based on this model, a total of 16 covariates were broadly categorised into four distinct groups (external environment, predisposing factors, enabling factors and need factors) (Figure 2). Geographical location of birth hospital (Cooma Health Service, Goulburn Base Hospital, Moruya District Hospital, Queanbeyan Health Service and South East Regional Hospital) and year of birth were categorised under external environment. Predisposing factors were maternal age at birth, maternal Aboriginal status and history of previous pregnancy. The index of relative socioeconomic advantage and disadvantage (IRSAD) was categorised under enabling factors. The construction of the IRSAD was based on area-specific economic and social conditions of people and households, including annual household equivalised income, employment classification, educational attainment and the types of occupied private dwellings and rental threshold. The details of measures included to construct the IRSAD are described elsewhere [19,20]. Need factors included maternal chronic hypertension, gestational diabetes, smoking during pregnancy, Apgar score, types of birth, low birthweight, sex of the baby and vaginal delivery (Figure 2).

### 2.5. Statistical Analysis

All the analyses were performed in STATA version 17 (College Station, TX 77845 USA). As part of the descriptive analyses, we first described the study population characteristics, followed by estimating exclusive breastfeeding on discharge rate with 95% confidence intervals (CIs). In the multivariable analyses, we adopted a hierarchical technique [20]. As part of the hierarchical technique, in the first stage, we entered variables from the external environment with a manual backward elimination process to remove non-significant variables (model 1). Significant variables in model 1 were analysed with all predisposing factors with a similar backward elimination process to remove non-significant variables in stage two (model 2). Similar processes were followed when enabling factors, need factors and maternity service use were included in the third (model 3), the fourth (model 4) and the fifth (model 5) stage, respectively. *p*-value < 0.05 was considered a significant level. Wald test was used to assess statistical significance, and variables significantly associated with 95% CI in the final model (model 5) were reported in the study. Collinearity was tested and reported.

### 2.6. Ethical Consideration

This study was approved by the Greater Western Human Research Ethics Committee (approval number: 2021/ETH10989, date of approval: 31 August 2021, protocol number: SNSWLHD 3_3_2021) and the Aboriginal Health and Medical Research Council Ethics Committee (approval number: 1998/22, date of approval: 26 September 2022, protocol number: SNSWLHD 3_3_2021). This study was authorised by the Southern New South Wales Local Health District (authorisation number: 2022/STE01605).

## 3. Results

### 3.1. Characteristics of Study Sample

From the sample of 13,168, the majority (27.5%) were from the Queanbeyan Health Service. Younger mothers and Aboriginal mothers constituted 4.4% and 6.4% of the sample, respectively (Table 1). A substantial percentage of the sampled women (60.8%) reported a previous pregnancy. Overall, 14.8% of women reported smoking during the second half of their pregnancy, whereas 2.9% of women who were smokers in the first half of pregnancy reported that they quit smoking in the second half of their pregnancy. Almost two-thirds of the sampled births were vaginal deliveries. Nearly 16% of women reported an inadequate number of antenatal care visits, whereas less than 50% women had their first ANC visit during the first trimester.

### 3.2. Exclusive Breastfeeding at Discharge Rate

Overall, the exclusive breastfeeding at discharge rate in the Southern NSW Local Health District for the period 2011–2020 was 84.2% (Table 1). The exclusive breastfeeding at discharge rate was as high as 86.3% in 2011, and as low as 82.0% in 2018 (Figure 3). The rate of exclusive breastfeeding at discharge differed significantly. Mothers aged 20 years or older, non-Aboriginal mothers, mothers having a higher index of socioeconomic advantage, non-smokers and those who had vaginal deliveries reported a higher rate of exclusive breastfeeding at discharge (Table 1).

### 3.3. Antenatal Care and Exclusive Breastfeeding at Discharge

Logistic regression analyses found that mothers who did not attend the recommended number of ANC visits were 15% less likely to exclusively breastfeed their infants at discharge (adjusted odds ratio (aOR) 0.85, 95% confidence interval (CI) 0.75, 0.97) compared to those who reported an adequate number of ANC visits (Table 2). Similarly, mothers who had their first ANC during the third trimester were 18% less likely to exclusively breastfeed their infants at discharge (adjusted odds ratio (aOR) 0.82, 95% confidence interval (CI) 0.68, 0.99) compared to those who had their first ANC during the first trimester (Table 2).

### 3.4. Factors Associated with Exclusive Breastfeeding at Discharge

The odds of exclusive breastfeeding at discharge among adolescent mothers was significantly lower compared to their non-teenage counterparts (20–34 years) (Table 3). Aboriginal mothers, mothers having a lower IRSAD score and mothers who reported gestational hypertension had a lower probability of exclusively breastfeeding at discharge, whereas non-smoker mothers had a higher probability of exclusively breastfeeding at discharge. Newborns with a lower Apgar score, multiple births and LBW infants were significantly less likely to be exclusively breastfed at discharge. In the final model, when the mother’s Aboriginal status was replaced by the infant’s Aboriginal status, the result indicates that Aboriginal infants were significantly less likely to be exclusively breastfed at discharge (aOR 0.78, 95% CI (0.66, 0.93)) compared to their non-Aboriginal peers (Table 3). With all birthing hospitals in the Southern NSW Local Health District providing similar levels of maternity care, Cooma Health Service was the referent category due to alphabetical default, consistent with our previous research [14].

## 4. Discussion

Increasing the number of infants exclusively fed with breast milk on discharge from the birth admission is one of the most important goals of breastfeeding policy in NSW [7]. Yet, exclusive breastfeeding rates on discharge in SNSWLHD decreased significantly for the period 2011–2020. Late ANC entry and failure to attend a recommended number of ANC visits were important predictors of a lower rate of exclusive breastfeeding on discharge. Predisposing factors described in our conceptual framework, such as young maternal age and mothers’ and infants’ Aboriginality status; enabling factors such as a lower IRSAD score; and need factors including gestational hypertension, a lower Apgar score, multiple births, low birthweight and caesarean birth were found to be associated with lower odds of exclusive breastfeeding on discharge.

In our study, we found concerning associations between lower rates of breastfeeding on discharge and families experiencing disadvantage. Our results demonstrate the size of the challenge ahead. Of particular concern is the inequity in breastfeeding outcomes, with younger mothers, Aboriginal mothers and mothers experiencing socioeconomic disadvantage being less likely to exclusively breastfeed on discharge.

To address these challenges, efforts must be made to increase access to quality midwifery care for all women. This could include strengthening the healthcare system and increasing the availability of trained midwives, as well as addressing cultural and societal barriers to seeking care. Additionally, interventions aimed at improving continuity of midwifery care, such as ensuring that the same midwife provides care throughout pregnancy, birth and postpartum, may be effective in improving breastfeeding outcomes [21].

A systematic comparison of models of care found that midwifery-led continuity models of care are associated with improved initiation and duration of breastfeeding, likely due to personalised and supportive care from a consistent provider, and fewer birth interventions [21]. Access to continuity models of midwifery care remains a challenge in many parts of the world, including Southern NSW. In some countries, midwifery care is not widely available or is fragmented, with different providers caring for women at different stages of pregnancy and postpartum. This fragmented care can lead to a lack of continuity and can negatively impact breastfeeding outcomes [21,22].

Southern NSW Local Health District’s birthing facilities serve a dispersed population, with women travelling up to 120 kilometres to attend their local birthing facility, with each facility servicing a similar catchment size. Rural caseload midwifery models offer a balance between access to localised pregnancy and postpartum care close to home, in a context of centralising birthing to larger facilities. Caseload midwifery requires an experienced and highly skilled workforce. In the context of widespread medical and midwifery workforce shortages, strategies to boost the rural midwifery workforce and opportunities to allow midwives to cultivate skills and work must be developed as a matter of urgency. Caseload midwifery models have been shown to be both feasible in rural settings and lead to higher workplace satisfaction for rural midwives [22,23,24].

Caseload midwifery is available to Aboriginal mothers and mothers of Aboriginal infants at Queanbeyan Health Service and Moruya District Hospital through the Aboriginal Maternal and Infant Health Service (AMIHS). In the AMIHS program, midwives and Aboriginal health workers work together to provide a continuity of care maternity service that is culturally safe, women-centred and meets the needs of Aboriginal families [3]. Our research found that these two sites have the highest proportion of women exclusively breastfeeding on discharge (Table 3), and we suggest that our findings tentatively support the expansion of the AMIHS program to other sites to improve uptake of ANC, and therefore birth and breastfeeding outcomes, among Aboriginal mothers and infants.

This study has several limitations. Because it is a cross-sectional study, we could not examine the causal relation of exposure and outcome variables, so although we have established an association between the timing and frequency of ANC and exclusive breastfeeding on discharge outcomes, we cannot exclude confounding variables. Further research is required to understand the elements of ANC visits that may contribute to improving breastfeeding initiation, particularly among identified populations of lower breastfeeding prevalence, including younger mothers, Aboriginal mothers and mothers experiencing socioeconomic disadvantage.

Our research was not able to examine exclusive breastfeeding beyond discharge due to a lack of comprehensive data collection. This study therefore makes an urgent call for future research to estimate exclusive breastfeeding rates and how these rates differ by priority population, including Aboriginal mothers. Understanding this along with the identification of local enablers and barriers through qualitative study may help to design evidence-informed breastfeeding interventions and close inequality in exclusive breastfeeding rates for priority populations.

However, our study also has several important strengths. The NSW Perinatal Data Collection includes every birth in NSW, regardless of setting, resulting in a highly representative sample of the Southern NSW Local Health District, without sampling bias. Our study demonstrated that pooling multiple years of data can contribute to longitudinal health policy evaluation in rural settings, where population data from single years or cohorts may be too limited to undertake useful statistical analysis.

The findings of this research may have implications for maternal and child health in Australia. By highlighting the factors impacting the promotion of exclusive breastfeeding through ANC, mothers and babies will benefit from the numerous health and nutritional benefits associated with this practice. This will contribute to the overall health and well-being of the Australian population and will support the country’s efforts to promote maternal and child health.

## 5. Conclusions

This research confirms that the timing and frequency of ANC visits are critical factors in determining breastfeeding outcomes for mothers and babies. With rates of exclusive breastfeeding on discharge declining in the Southern NSW Local Health District in the previous decade, this research serves as a call to action to address the barriers that prevent women from engaging in ANC visits.

By providing regional and rural mothers with important information, resources and support through coordinated, continuous ANC models of care, we can improve breastfeeding outcomes and promote the health and well-being of both mothers and infants. Further research is needed to better understand the relationship between ANC visits and breastfeeding outcomes and to develop strategies for increasing the frequency of these visits.

To address these challenges, efforts must be made to increase access to quality ANC services for all women, but particularly Aboriginal mothers and infants, younger mothers and mothers experiencing disadvantage. This may include strengthening the healthcare system through expansion of caseload midwifery models of care, tailored ANC programs and increasing the midwifery workforce, as well as addressing cultural and societal barriers to seeking care. Additionally, interventions aimed at increasing the frequency of ANC visits, such as providing transportation or financial incentives, may be effective in improving breastfeeding outcomes.

## Figures and Tables

**Figure 1 ijerph-20-06135-f001:**
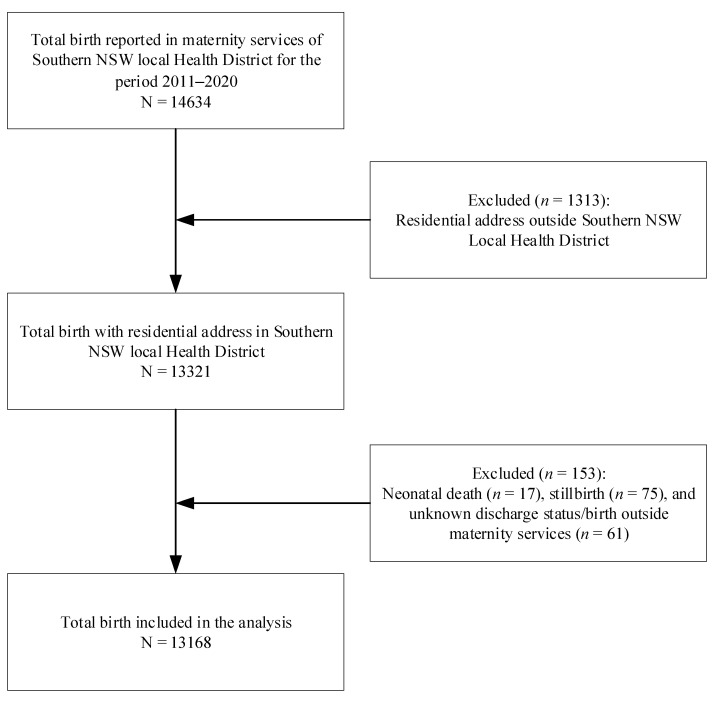
Flow diagram of sample selection for exclusive breastfeeding at discharge in SNSWLHD (2011–2020).

**Figure 2 ijerph-20-06135-f002:**
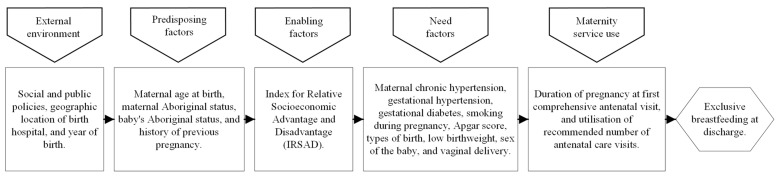
Conceptual framework for the use of maternity service and exclusive breastfeeding in SNSWLHD, adopted from Anderson’s behavioural model.

**Figure 3 ijerph-20-06135-f003:**
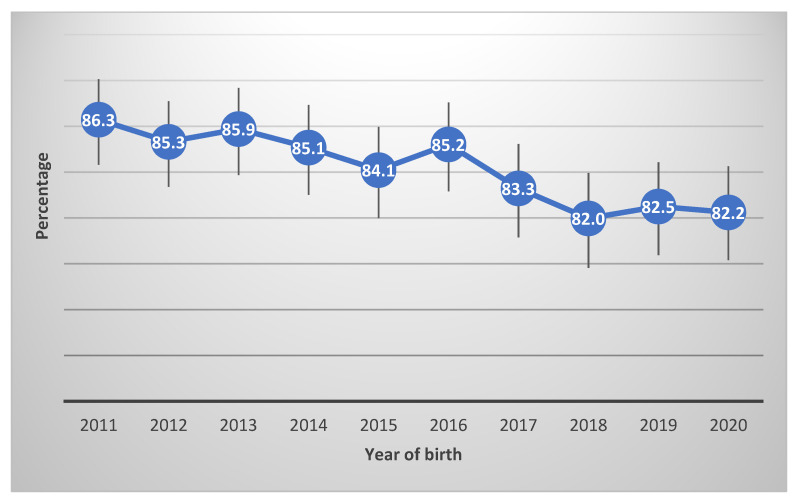
Trend in exclusive breastfeeding at discharge rate with 95% CI in SNSWLHD (N = 13,168).

**Table 1 ijerph-20-06135-t001:** Characteristics of study population, exclusive breastfeeding at discharge numbers and rates with 95% CI in SNSWLHD (2011–2020).

Study Factors	Study Sample	Exclusive Breastfeeding at Discharge	
Covariates	N (% ^ϕ^)	*n*	% (95% CI)
Geographic location of birth hospital			
Cooma Health Service	1479 (11.2)	1241	83.9 (81.9, 85.7)
Goulburn Base Hospital	2904 (22.1)	2298	79.1 (77.6, 80.6)
Moruya District Hospital	2813 (21.4)	2433	86.5 (85.2, 87.7)
Queanbeyan Health Service	3624 (27.5)	3182	87.8 (86.7, 88.8)
South East Regional Hospital	2348 (17.8)	1931	82.2 (80.6, 83.7)
Maternal age at birth			
<20 years	582 (4.4)	429	73.7 (70.0, 77.1)
20–34 years	10,416 (79.1)	8831	84.8 (84.1, 85.5)
35+ years	2170 (16.5)	1825	84.1 (82.5, 85.6)
Maternal Aboriginal status (N = 13,131)			
Non-Aboriginal	12,291 (93.3)	10,419	84.8 (84.1, 85.4)
Aboriginal	840 (6.4)	636	75.7 (72.7, 78.5)
Baby’s Aboriginal status (N = 12,944)			
Non-Aboriginal	11,803 (89.6)	10,002	84.7 (84.1, 85.4)
Aboriginal	1141 (8.7)	890	78.0 (75.5, 80.3
History of previous pregnancy (N = 13,152)			
Yes	8000 (60.8)	6760	84.5 (83.7, 85.3)
No	5152 (39.1)	4311	83.7 (82.6, 84.7)
IRSAD quintile (N = 13,154)			
First (most socially advantaged)	1584 (12.0)	1401	88.4 (86.8, 89.9)
Second	864 (6.6)	763	88.3 (86.0, 90.3)
Third	2512 (19.1)	2166	86.2 (84.8, 87.5)
Fourth	5029 (38.2)	4239	84.3 (83.3, 85.3)
Fifth (most socially disadvantaged)	3165 (24.0)	2505	79.1 (77.7, 80.5)
Chronic hypertension			
No	13,059 (99.2)	10,997	84.2 (83.6, 84.8)
Yes	109 (0.8)	88	80.7 (72.2, 87.1)
Gestational hypertension			
No	12,775 (97.0)	10,784	84.4 (83.8, 85.0)
Yes	393 (3.0)	301	76.6 (72.1, 80.5)
Gestational diabetes (N = 13,147)			
No	12,348 (93.8)	10,413	84.3 (83.7, 85.0)
Yes	799 (6.0)	654	81.9 (79.0, 84.4)
Smoking during pregnancy (N = 13,130)			
Smoker during the second half of pregnancy	1951 (14.8)	1415	72.5 (70.5, 74.5)
Smoker in the first half of pregnancy but quit in the second half	385 (2.9)	297	77.1 (72.7, 81.1)
Non-smoker	10,794 (82.0)	9342	86.5 (85.9, 87.2)
Apgar 5 min score (N = 13,068)			
Apgar ≥ 7	12,887 (97.9)	10,889	84.5 (83.9, 85.1)
Apgar < 7	181(1.4)	111	61.3 (54.0, 68.1)
Types of birth			
Singleton	13,066 (99.2)	11,041	84.5 (83.9, 85.1)
Twin	102 (0.8)	44	43.1 (33.9, 52.9)
Low birthweight (<2500 g)			
No	12,788 (97.0)	10,862	84.9 (84.3, 85.5)
Yes	380 (3.0)	223	58.7 (53.7, 63.5)
Sex of the baby (N = 13,166)			
Male	6771 (51.4)	5665	83.7 (82.8, 84.5)
Female	6395 (48.6)	5419	84.7 (83.8, 85.6)
Vaginal delivery			
Yes	9620 (73.1)	8289	86.2 (85.5, 86.8)
No	3548 (26.9)	2796	78.8 (77.4, 80.1)
Exposure variables			
Utilisation of recommended number of ANC visits			
Yes	11,114 (84.4)	9439	84.9 (84.3, 85.6)
No	2054 (15.6)	1646	80.1 (78.4, 81.8)
Duration of pregnancy at first comprehensive antenatal visit (N = 13,120)			
First trimester	6198 (47.1)	5225	84.3 (83.4, 85.2)
Second trimester	5921 (45.0)	5010	84.6 (87.6, 85.5)
Third trimester	1001 (7.6)	818	81.7 (79.2, 84.0)
Total	13,168 (100)	11,085	84.2 (83.5, 84.8)

^ϕ^ Column percentage does not add up to 100 because of missing values. IRSAD: index of relative socioeconomic advantage and disadvantage; ANC: antenatal care; CI: confidence interval.

**Table 2 ijerph-20-06135-t002:** Impact of timing and number of antenatal care visits on exclusive breastfeeding at discharge rates in SNSWLHD (2011–2020).

	Crude		Adjusted ^£^	
Exposure Variables	OR	(95% CI)	n	% (95% CI)
Utilisation of recommended number of antenatal care visits				
Yes	Reference		Reference	
No	0.72	(0.63, 0.81) **	0.85	(0.75, 0.97) *
Duration of pregnancy at first comprehensive antenatal visit				
First trimester	Reference		Reference	
Second trimester	1.02	(0.93, 1.13)	0.93	(0.84, 1.04)
Third trimester	0.79	(0.67, 0.94) *	0.82	(0.68, 0.99) *

**: *p* < 0.001; *: *p* < 0.05; ORs: odds ratios; CI: confidence interval; ^£^ adjusted for: year of birth, geographic location of birth hospital, maternal age at birth, maternal Aboriginal status, baby’s Aboriginal status, history of previous pregnancy, wealth quintile, chronic hypertension, gestational hypertension, gestational diabetes, smoking during pregnancy, Apgar 5 min score, birthweight, sex of the baby and vaginal delivery.

**Table 3 ijerph-20-06135-t003:** Factors associated with exclusive breastfeeding at discharge in SNSWLHD (2011–2020).

Covariates	Crude		Adjusted ^£^	
	OR	(95% CI)	n	% (95% CI)
Year of birth				
2011	Reference		Reference	
2012	0.92	(0.74, 1.15)	0.98	(0.77, 1.23)
2013	0.97	(0.77, 1.21)	0.95	(0.75, 1.21)
2014	0.91	(0.73, 1.13)	0.91	(0.72, 1.15)
2015	0.84	(0.68, 1.04)	0.84	(0.67, 1.06)
2016	0.92	(0.73, 1.14)	0.90	(0.71, 1.14)
2017	0.79	(0.64, 0.98) *	0.72	(0.57, 0.90) *
2018	0.72	(0.59, 0.89) *	0.68	(0.54, 0.85) *
2019	0.75	(0.61, 0.92) *	0.71	(0.56, 0.89) *
2020	0.74	(0.60, 0.91) *	0.68	(0.54, 0.85) *
Geographic location of birth hospital				
Cooma Health Service	Reference		Reference	
Goulburn Base Hospital	0.73	(0.62, 0.86) **	0.82	(0.64, 1.06)
Moruya District Hospital	1.23	1.23 (1.03, 1.46) *	0.84	(0.68, 1.04)
Queanbeyan Health Service	1.38	1.38 (1.16, 1.64) **	1.48	(1.15, 1.90) *
South East Regional Hospital	0.89	0.89 (0.75, 1.06)	0.93	(0.71, 1.20)
Maternal age at birth				
20–34 years	Reference		Reference	
<20 years	0.50	(0.42, 0.61) **	0.60	(0.48, 0.75) **
35+ years	0.95	(0.84, 1.08)	0.93	(0.81, 1.07)
Maternal Aboriginal status				
Non-Aboriginal	Reference		Reference	
Aboriginal	0.56	(0.47, 0.66) **	0.72	(0.60, 0.87) *
Baby’s Aboriginal status				
Non-Aboriginal	Reference		Reference	
Aboriginal	0.64	(0.55, 0.74)	0.78	(0.66, 0.93) *
IRSAD quintile				
First (most socially advantaged)	Reference		Reference	
Second	0.99	(0.76, 1.28)	1.05	(0.78, 1.43)
Third	0.82	(0.68, 0.99) *	0.92	(0.75, 1.13)
Fourth	0.70	(0.59, 0.83) **	0.74	(0.56, 0.98) *
Fifth (most socially disadvantaged)	0.50	(0.42, 0.59) **	0.63	(0.48, 0.82) *
Gestational hypertension				
No	Reference		Reference	
Yes	0.60	(0.48, 0.77) **	0.60	(0.47, 0.77) **
Smoking during pregnancy				
Smoker at the end of pregnancy	Reference		Reference	
Smoker in the first half of pregnancy but quit in the second half	1.28	(0.99, 1.65)	1.21	(0.92, 1.58)
Non-smoker	2.44	(2.18, 2.73) **	2.08	(1.83, 2.37) **
Apgar 5 min score				
Apgar ≥ 7	Reference		Reference	
Apgar < 7	0.29	(0.21, 0.39) **	0.35	(0.25, 0.49) **
Types of birth				
Singletons	Reference		Reference	
Twin	0.14	(0.09, 0.21) **	0.23	(0.15, 0.36) **
Low birthweight (<2500 g)				
No	Reference		Reference	
Yes	0.25	(0.20, 0.31) **	0.46	(0.36, 0.59) **
Vaginal delivery				
Yes	Reference		Reference	
No	0.60	(0.54, 0.66) **	0.59	(0.53, 0.66) **

**: *p* < 0.001; *: *p* < 0.05; ORs: odds ratios; CI: confidence interval; ^£^ adjusted for: year of birth, geographic location of birth hospital, maternal age at birth, maternal Aboriginal status, baby’s Aboriginal status, history of previous pregnancy, wealth quintile, chronic hypertension, gestational hypertension, gestational diabetes, smoking during pregnancy, Apgar 5 min score, birthweight, sex of the baby, vaginal delivery and utilisation of recommended antenatal care visits.

## Data Availability

Access to the data used in this study is in accordance with the research protocol submitted to the Human Research Ethics Committees. For data inquiries, please contact the Greater Western Human Research Ethics Committee (GWHREC) (postal address: P.O. Box 143, 39 Hampden Park Road, Bathurst, Australia 2795; Tel.: +61-02-6330-5948).

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
