# Peer review of "Exclusive Breastfeeding at Discharge in Regional New South Wales, Australia: The Role of Antenatal Care (2011–2020)"

_ijerph, 2023, doi:10.3390/ijerph20126135_

Round 1
Reviewer 1 Report
Reviewer’s comments - IJERH 2363937
Title: Exclusive breastfeeding at discharge in regional New South Wales, Australia: The role of antenatal care (2011-2020).
Comment to Authors:
Thank you for your interesting and important work on the role of antenatal care and that factors that contribute to exclusive breastfeeding.
The description of the study population, the analysis of the timing and number of the ANC on exclusive breastfeeding and the analysis of factors associated with exclusive breastfeeding is appropriate. The questions that remains unclear is what is the link between ANC visit and factors that are associated with exclusive breastfeeding? Can you link ANC visits with geographic location, maternal age, aboriginal status etc. This would potentially strengthen your arguement that ANC visits are play a pivotal role in supporting exclusive breastfeeding.
Comments
1. Abstract
In the abstract you state that breastfeeding areas have declined over the past decade (line 16). Line 20 you say that you have showed exclusive breastfeeding rates in SNSWLHD have declined – can you add a short statement here to acknowledge that your findings are in alignment with previous literature. It will help this section to not feel like your findings are already well known within the literature.
2. Introduction:
Line 39-44 and 49-58: You introduce the concept of antenatal visits. How do you known that time is allocated during the antenatal visits to breastfeeding education/support? And is this consistent across different clinicians/health services? Are there any policy/guidelines around the structure of the antenatal visits that ensures breastfeeding education and support is part of the antenatal visit? Or is any breastfeeding support/education from antennal visit dependent on the clinician and/or time available in the appointment (i.e. less time available for breastfeeding education/support in a complicated pregnancy?).
Line 47-48: Notes a drop in exclusive breastfeeding rates at discharge between 2011 and 2020. In 2020 there was a significant disruption to healthcare delivery due to the pandemic. What impact has this had on these declining rates?
Line 58: Check grammar.
3. Materials and Methods
Line 118: Check grammar.
4. Results
Section 3.3: Please reference table 2.
Table 3: Why was Cooma Health Service used as the referent?
5. Discussion
Line 198-22 and 207-209 are repetitive. Suggest include only once.
Line 211-212: You note that your results demonstrate the ‘urgency’ of the problem. The ‘urgency’ of the problem is not a clear connection with the results.
Line 212-213: The connection between ANC and the factors linked to exclusive breastfeeding outcomes is not clear. See general comments.
Line 220: Can you please refence the statement made here.
Line 229: Check grammar.
Line 229-235: You introduce a discussion on rurality. You have not presented any information within your results about the rurality of your health service catchments. Data needs to be given (suggest using the Modified Monash Model or similar) on the rurality of each of your health services and the differences between their population catchments.
Line 246: You mention “exposure and outcome variables.” This limitation is generic to all cross-sectional studies, can you add specificity to this sentence to link the limitation more directly to your data?
6. Reference
Check formatting of references 11, 12, 13. Journal name not italicised.
English language is appropriate. Minor grammatical changes.
Reviewer 2 Report
This is an important piece of work that demonstrates the unacceptable decline in exclusive BF rates within NSW Australia and how the health system is failing vulnerable mothers to provide the best nutrition to their child at the start of life. This is horrifying data and an important piece of work to generate change in the health care system.
Overall, the paper is very area specific, which is important to the study design but throughout the paper there needs to be explanation and comparison of this to the global setting. The readership of the journal in global and the authors need to provide more context around where NSW is in Australia. How this is one of 8 states and territories in Australia and has what population size? The rural region discussed also needs to be given a clearing picture of size and demographics to the global reader. Discussion around the term first nations peoples in Australia is also needed to provide context to the findings relating Aboriginal mothers. The context of poorer health outcomes for ingenious peoples not only here in Australia but also in places like Canada is needed to provide global context for the paper.
Introduction
More information needs to be provided for the global reader about how the health cares system is delivered in Australia and why and how free ANC is available. Many readers may be from Low Middle-Income countries and are unaware of how antenatal care is delivered state by state in Australia. In addition, more information around what occurs in the ANC visit is needed. E.g. is BF advice provided or general health check up of the mother? What is so important about these visits that can help EBF on discharge? Also, who are the ANC delivered by? GP’s Hospital staff? In hospitals or community care? More contextual information required.
Line 60- Southern NSW Local Health District?? This would only be known by a very small % of Australian readers. Make sure this is explained to global readership
Methods
Line 71-79 needs more information for the global reader – what is NSW e.g. a state within Australia with XX population. For each state in Australia population health data is collected via….
Line 82 Outcome Variable? Why no 6mth EBF rate data? A 6 moth rate for this cohort would have added important data to the findings? The reason why this was not included need to be discussed in both methods and limitations.
112 Framework? Unsure why the framework is included. As it stands there is no linking explanation of the formwork to how the variables are tested and then presented in the results. Without integration of the framework throughout methods, results and discussion this becomes tokenistic. I would either integrating more or removing the figure.
Results
Paragraph 3.3 and 3.4 need to direct the reader to the table below for the data points discussed.
Discussion
Overall, the discussion needs more integration of supporting evidence and citations from both local and global perspectives. Many sentences are stated without discussion from supporting evidence e.g. line 215-220. The statements in this paragraph required supporting evidence and discussion around what ‘health systems’ are they talking about. Those in High income countries or those in LIC as there are vast differences here. Explanation with examples for other global settings such as UK, Ireland with a similar health care system to Australia would be good.
What level of health care strengthening is needed? At the local primary care level, hospital midwifery programs, community care programs? More integration of discussion around these different elements of care needed.
Line 242 – was this reported in the results? I couldn’t find this reported. If not reported in results, then you cannot refer to the findings here. If research is from another study, you must cite the publication the results are from. To include this either you need to add these findings into the results more clearly or cite the article.
Given that one of the major findings was that Aboriginal and young mums and poorer EBF on discharge. More detailed discussion is needed around culturally safe and inclusion models of ANC it is discussed briefly above but what have other countries with ingenious populations done (e.g. Canada, Norway) wo increase EBF rates within their indigenous populations? Powerful role of indigenous midwives or birthing elders?
Young mums and how better to engage with these vulnerable mums is also needed. Is a greater level of support and care that include not only the mother but also a ‘mentor’ needed?
Limitations
State why no 6mth follow up EBF rate not included?
Conclusion
Line 280 – trained in what? More specific detail here is this training in ANC or cultural competency? This is the first mention of training. This would be better included in the discussion
Reviewer 3 Report
This is well written paper. Please correct exclusive breastfeeding rates given in the text (80.2) and in the Figure 3 (82.2) for the year 2020.
Author Response
Thank you for the opportunity to provide a response to your comment on our paper on ‘Exclusive breastfeeding at discharge in regional New South Wales, Australia: The role of antenatal care (2011-2020)’. We appreciate your time in viewing our paper and your valuable comment.
The manuscript has been revised incorporating the feedback from reviewers. We have addressed the error that you spotted, and have corrected it in our revised manuscript (Line 206).
Round 2
Reviewer 1 Report
Thank you for addressing the comments.
Can you fix the minor error in the references - there is no reference listed under reference 10.
Author Response
Thank you for spotting this error. The manuscript has been updated, with the reference at Line 386, and in-text reference at Line 89.
